# A Novel Complex-Valued Blind Source Separation and Its Applications in Integrated Reception

Weilin Luo [ID], Hongbin Jin, Xiaobai Li *, Hao Li, Kang Liu and Ruijuan Yang

Department of Intelligence, Air Force Early Warning Academy, Wuhan 430019, China; wlinluo@163.com (W.L.); jhb760817@sina.com (H.J.); afeu_li@163.com (H.L.); 18504229299@163.com (K.L.); ruijuany@sohu.com (R.Y.)
* Correspondence: lxb2cici@163.com

**Abstract:** The separation of time–frequency mixing signals composed of radar, communication, and jamming is the first step in integrated reception processing, which requires higher accuracy for complex blind source separation (CVBSS). However, traditional CVBSS methods have limitations such as low separation accuracy, a slow convergence speed, and poor robustness in low signal-to-noise ratio (SNR) and high jamming-to-signal ratio (JSR) scenarios. To address the above issues, this paper firstly establishes a time delay mixing mathematical model. A robust whitening algorithm is proposed by using the time delay correlation matrix of the observed signal, which is insensitive to noise. Secondly, the joint diagonalized F-parametrization is used as the objective function, and the separation matrix is constructed based on the multiple complex-valued Givens matrices. The complex-valued Givens matrix not only ensures orthogonality in the separation matrix but also effectively reduces the number of parameters to be calculated. This approach guarantees accuracy and simplifies the complexity of the separation process. Finally, the nonlinear chaotic grey wolf optimizer is utilized to search for the optimal rotation angle. The simulation results demonstrate that this algorithm offers higher separation accuracy and requires fewer iterations compared to the traditional algorithm. Additionally, it enhances the accuracy of direction of arrival (DOA) estimation, reduces the communication bit error rate, and enables the joint estimation of the target distance and velocity even in the presence of powerful jamming and a low SNR.

**Keywords:** integrated reception; complex-valued blind source separation; robust whitening; joint diagonalization; anti-main lobe jamming; swarm intelligence



## 1. Introduction

In the complex electromagnetic environment, numerous radar, communication, jamming, and other radiation signals, whether cooperative or non-cooperative, are present. The integrated receiver's wide beam and wide bandwidth coverage often result in overlapping signals in the time–frequency and air domains. Consequently, performing tasks like parameter estimation and information extraction becomes challenging due to the low signal-to-noise ratio (SNR) and high jamming-to-signal ratio (JSR). Under conditions of low SNR, valuable signals may become obscured by strong noise, making it challenging to detect radar signals and increasing the likelihood of bit errors in communication signals. Conversely, if the jamming signal is too strong, jamming can suppress the useful signal, resulting in the failure of or errors in radar signal detection and the inability to demodulate the communication signals. Therefore, the primary concern is to separate these overlapping signals. Blind source separation (BSS) [1,2] aims to recover or reconstruct the source signal using observed data from the receiver channel, even when the source signal and mixing characteristics are unknown. BSS, as a powerful technology in signal processing, has played a crucial role in various domains, including speech separation [3–5], wireless communication [6–8], anti-main lobe jamming [9–13], and direction of arrival (DOA) estimation [14,15]. BSS has also provided innovative ideas for ongoing research in integrated

reception processing. In this paper, a complex scenario of integrated reception is considered, as depicted in Figure 1. In this scenario, radar signals, echo signals, jamming signals, and communication signals overlap in the time–frequency domain. After separating the mixed signals, the mixing matrix is used to estimate the DOA. The echo signals undergo pulsed Doppler processing, the communication signal is demodulated, reconnaissance signals are sorted, and the jamming signal is analyzed. This method enables the execution of tasks such as location, detection, reconnaissance, communication, and anti-jamming.

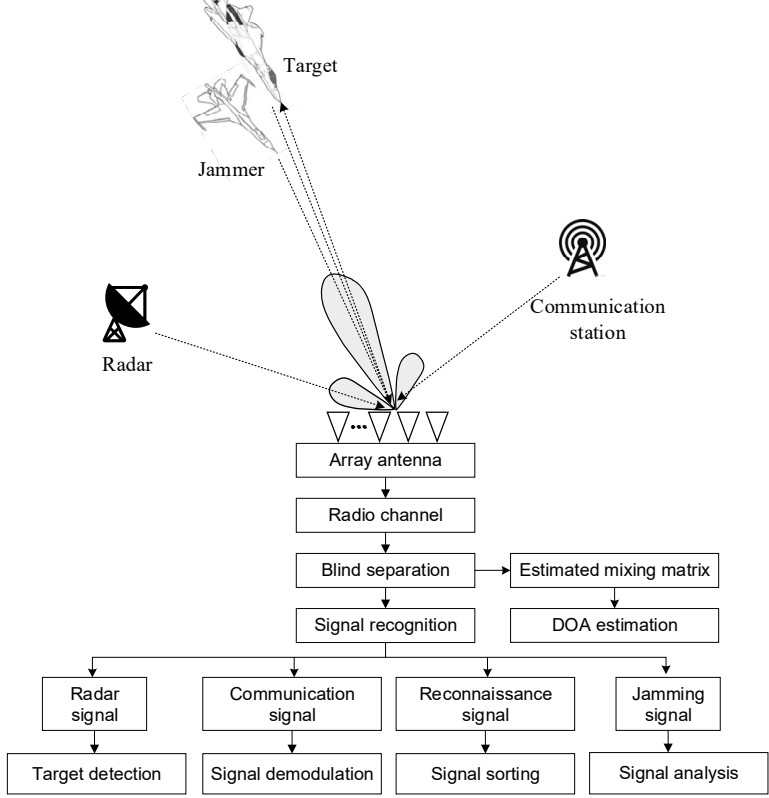

**Figure 1.** A type of integrated reception system.

When using arrays to receive far-field narrowband signals, which are part of the time delay mixing mathematical model [16], it is necessary to employ complex-valued blind source separation (CVBSS). However, traditional CVBSS algorithms suffer from low separation accuracy, slow convergence, poor robustness, and high sensitivity to noise and jamming. The well-known second-order blind identification (Sobi) [17,18] is a classical and effective algorithm to solve BSS problems using second-order statistics. Although it has low computational complexity, it exhibits poor separation performance. In [19], the joint approximate diagonalization of eigenmatrices (JADE) algorithm was proposed. This algorithm tackles the objective function maximization problem by jointly diagonalizing the eigenmatrices of a set of fourth-order cumulant matrices, thereby significantly improving the separation performance. The computational complexity is high due to the fourth-order cumulant matrix. To address this, c-FastICA [20] extends the FastICA algorithm from the real domain to the complex-valued domain. However, c-FastICA is limited to circular signals. To handle non-circular signals, Novey et al. developed the nc-FastICA algorithm [21]. Another approach is the RobustICA algorithm proposed in the literature [22,23], which combines an exact linear search with a cost function called cliquishness. RobustICA addresses the convergence issue of c-FastICA and nc-FastICA, which may lead to a saddle point. However, RobustICA uses cliquishness as the objective function and is more sensitive to noise and outliers. Another algorithm, FAJD [24], is a non-orthogonal joint diagonalization algorithm that avoids degenerate solutions by introducing a penalty

term to the non-diagonal error criterion equation. However, the penalty term increases the computational complexity of the algorithm.

The algorithms mentioned above all utilize the zero time delay autocorrelation matrix of the observed signal for whitening. However, it is important to note that the zero time delay autocorrelation matrix does not eliminate the impact of additive noise, leading to lower separation accuracy in low SNR and high JSR scenarios. Additionally, most of these algorithms rely on the Newton optimizer, which is highly sensitive to the initial value and prone to becoming stuck in saddle points. In recent years, swarm intelligence-based BSS objective functions such as kurtosis and negative entropy have gained attention [25–29]. However, these approaches primarily focus on real-valued BSS through instantaneous mixing models, and they rarely consider swarm intelligence as a joint diagonalization optimization algorithm. To address these concerns, this paper introduces a time delay mixing model and proposes a method to correct amplitude and phase errors, ensuring the accurate demodulation of the communication signal. Secondly, the whitening matrix is constructed using multiple time delay correlation matrices of the observed signals, which enhances its anti-noise ability due to the correlation matrix of Gaussian noise being 0. Additionally, the separation matrix is constructed using multiple complex-valued Givens matrices. These matrices not only satisfy the orthogonality constraint of the separation matrix but also effectively reduce the number of parameters to be calculated, simplifying the separation complexity while ensuring accuracy. To further enhance the performance, a nonlinear convergence factor and chaotic perturbation are incorporated into the standard grey wolf optimizer (GWO), resulting in the nonlinear chaotic grey wolf optimizer (NCGWO). Simulation results demonstrate that the proposed algorithm exhibits higher separation accuracy and requires fewer iterations, particularly under low SNR and high JSR conditions. These findings lay a solid foundation for applications such as anti-main lobe jamming, the reduction of the bit error rate (BER), and the joint estimation of the DOA, distance, and velocity. The main contributions or novelties of this paper are as follows.

(1) This paper first proposes an integrated receiving system that is based on signal separation. The system utilizes a time delay mixed model, allowing it to receive radar, communication, and jamming signals that suffer from time–frequency aliasing. By applying a blind source separation algorithm, the mixed signals can be effectively separated. This enables the system to perform various tasks such as target position and velocity estimation, the extraction of communication information, anti-interference measures, and direction of arrival (DOA) estimation.

(2) A robust whitening method is employed instead of the traditional method. This method utilizes multiple time delay covariance matrices of observed signals to construct a whitening matrix, effectively suppressing the influence of noise. Additionally, we introduce a new method to accurately calculate the coefficients of the delay covariance matrix.

(3) The F-norm of the joint diagonalization is used as the objective function. This replaces kurtosis and negative entropy, which are traditionally used in swarm intelligence in solving the blind source separation. Unlike kurtosis and negative entropy, the F-norm does not require multiple averaging operations, making it more efficient when dealing with large amounts of data. The F-norm solely relies on matrix operations, which are known for their fast computation speed.

(4) This paper proposes a nonlinear chaotic grey wolf optimizer to solve the objective function. In order to optimize the performance, we apply a restriction on the unitary matrix using multiple complex Givens matrices. This restriction not only reduces the number of parameters in the solution but also accelerates the overall solution process.

## 2. Time Delay Mixing Model

The linear time delay mixing model considers both the time delay and amplitude attenuation of the source signal arriving at each sensor. Considering that there are $K$ independent far-field narrowband signals incident on an $M$-element uniform line array

from different directions $\theta_1, \theta_2, \cdots, \theta_K$, if the $k$-th target signal received by the first array element is $s_k(t)$, the mixing signal received by the $m$-th array element is [30]

$$x_m(t) = \sum_{k=1}^{K} s_k(t) \exp(-j2\pi f_k \tau_{mk}) + n_m(t)(1 \le m \le M, 1 \le k \le K) \tag{1}$$

where $t = t_1, t_2, \cdots, t_N$ is the time series, $N$ is the array snapshot number, $f_k$ is the carrier frequency of the target signal, $\tau_{mk}$ is the time delay of the $k$-th signal reaching the $m$-th array element, and $n_m(t)$ is the output noise of the $m$-th array element.

As shown in Figure 2, for a uniform linear array, the time delay can be expressed as $\tau_{mk} = \frac{d \sin \theta_k}{c}$ based on the first array element. $d$ is the distance between adjacent elements, and $c$ is the speed of light. The output of all array elements is expressed in matrix form as

$$x(t) = As(t) + n(t) \tag{2}$$

where the observed signal is $x(t) = [x_1(t), x_2(t), \ldots, x_M(t)]^T$, the source signal is $s(t) = [s_1(t), s_2(t), \ldots, s_K(t)]^T$, and the noise is $n(t) = [n_1(t), n_2(t), \ldots, n_M(t)]^T$, $M \ge K$. $n_m(t)$ is complex-valued Gaussian white noise with zero mean. The output noise of each array element is independent of each other. The array matrix (mixing matrix) $A$ is

$$A = \left\{ \begin{array}{cccc} 1 & 1 & \cdots & 1 \\ e^{-j2\pi d \frac{\sin\theta_1}{\lambda}} & e^{-j2\pi d \frac{\sin\theta_2}{\lambda}} & \cdots & e^{-j2\pi d \frac{\sin\theta_K}{c}} \\ \vdots & \vdots & \vdots & \vdots \\ e^{-j2\pi(M-1)d \frac{\sin\theta_1}{\lambda}} & e^{-j2\pi(M-1)d \frac{\sin\theta_2}{\lambda}} & \cdots & e^{-j2\pi(M-1)d \frac{\sin\theta_K}{\lambda}} \end{array} \right\} \tag{3}$$

where $\lambda$ is the signal wavelength. It is obvious that the array flow pattern matrix $A$ contains the DOA of the signal.

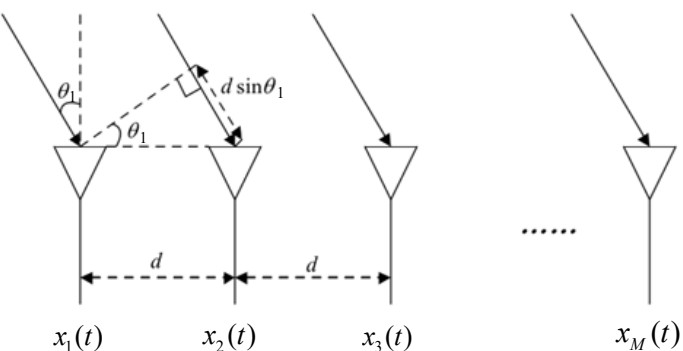

**Figure 2.** Far-field narrowband signal mixing model.

The purpose of BSS is to obtain the separation matrix $B$ by some algorithm, so that

$$y(t) = Bx(t) \tag{4}$$

where $y(t)$ is the separated signal (estimation of the source signal). CVBSS generally has uncertainty in the amplitude, phase, and sequence. Amplitude ambiguity affects signal detection and phase ambiguity affects signal demodulation in an integrated system. Therefore, amplitude and phase correction are needed.

Without considering noise, if the array is ideal, the separation matrix $B$ and the mixing matrix $A$ have the following relationship.

$$BA = DP \tag{5}$$

where $P$ is the permutation matrix of dimension $K \times K$, which represents the sequence ambiguity of CVBSS. $D = diag\{\alpha_1, \alpha_2, \cdots, \alpha_K\}$ is the diagonal matrix, which represents the amplitude and phase ambiguity of CVBSS. $\alpha_k \in \mathbb{C}$ denotes the amplitude phase ambiguity between the $i$-th estimated signal and its corresponding source signal. When there is no amplitude phase ambiguity in this signal component, $\alpha_i = 1$. $\hat{A}$ is the mixing matrix estimated using the CVBSS. Then, $\hat{A}$ is the left pseudo-inverse matrix of $B$, i.e., $\hat{A} = (B^H B)^{-1} B^H$. Replacing it into Equation (5),

$$\hat{A} = AP^{-1}D^{-1} \tag{6}$$

Since the inverse of the permutation matrix $P^{-1}$ is still a permutation matrix, $AP^{-1}$ is swapping the positions of some columns in $A$. Let the left-to-right columns of $A$ be numbered $1, 2, \cdots, K$. The numbering of the replacement is $P_1, P_2, \cdots, P_K, \phi_k = 2\pi d \frac{\sin \theta_k}{\lambda}$. Then, $AP^{-1}$ can be expressed as

$$AP^{-1} = \begin{bmatrix} 1 & 1 & \cdots & 1 \\ e^{-j\phi_{P_1}} & e^{-j\phi_{P_2}} & \cdots & e^{-j\phi_{P_K}} \\ \vdots & \vdots & \ddots & \vdots \\ e^{-j(M-1)\phi_{P_1}} & e^{-j(M-1)\phi_{P_2}} & \cdots & e^{-j(M-1)\phi_{P_K}} \end{bmatrix} \tag{7}$$

Substituting $AP^{-1}$ and $D$ into Equation (6),

$$\begin{aligned} \hat{A} &= \begin{bmatrix} 1 & 1 & \cdots & 1 \\ e^{-j\phi_{P_1}} & e^{-j\phi_{P_2}} & \cdots & e^{-j\phi_{P_K}} \\ \vdots & \vdots & \ddots & \vdots \\ e^{-j(M-1)\phi_{P_1}} & e^{-j(M-1)\phi_{P_2}} & \cdots & e^{-j(M-1)\phi_{P_K}} \end{bmatrix} \begin{bmatrix} \alpha_1^{-1} & & & \\ & \alpha_2^{-1} & & \\ & & \ddots & \\ & & & \alpha_K^{-1} \end{bmatrix} \\ &= \begin{bmatrix} \alpha_1^{-1} & \alpha_2^{-1} & \cdots & \alpha_K^{-1} \\ \alpha_1^{-1} e^{-j\phi_{P_1}} & \alpha_2^{-1} e^{-j\phi_{P_2}} & \cdots & \alpha_K^{-1} e^{-j\phi_{P_K}} \\ \vdots & \vdots & \ddots & \vdots \\ \alpha_1^{-1} e^{-j(M-1)\phi_{P_1}} & \alpha_2^{-1} e^{-j(M-1)\phi_{P_2}} & \cdots & \alpha_K^{-1} e^{-j(M-1)\phi_{P_K}} \end{bmatrix} \end{aligned} \tag{8}$$

It can be seen that the elements of the first row of $\hat{A}$ contain exactly the amplitude phase ambiguity information of each signal. The amplitude phase ambiguity can be extracted from $\hat{A}$ to correct the estimated signal for amplitude phase. Thus,

$$D^{-1} = diag\{\hat{A}(1, :)\} \tag{9}$$

After considering the noise, the output obtained by CVBSS is

$$\begin{aligned} \mathbf{y}(t) &= BAs(t) + Bn(t) \\ &= DPs(t) + \Delta s(t) \end{aligned} \tag{10}$$

where $\Delta s(t) = Bn(t)$ denotes the error due to additive noise. Therefore, the amplitude phase correction for CVBSS can be performed by left multiplying the matrix $y(t)$ by $D^{-1}$, i.e.,

$$\hat{s}(t) = D^{-1}y(t) \tag{11}$$

Substituting $y(t)$ into Equation (11),

$$\hat{s}(t) = Ps(t) + D^{-1}\Delta s(t) \tag{12}$$

It can be seen that the error $\Delta s(t)$ still exists between $\hat{s}(t)$ and $s(t)$ due to the sequence ambiguity $P$ and additive noise. From the expression of $A$, for each column of $A$, the DOA corresponding to a source signal is obtained by

$$\hat{\theta}_k = \frac{1}{M-1} \sum_{i=1}^{M-1} \arcsin\left(-\frac{\lambda \arg\{\hat{a}_k(i+1)/\hat{a}_k(i)\}}{2\pi d}\right) \tag{13}$$

$\hat{a}_k(i)$ represents the *i*-th element in column *k* of $\boldsymbol{A}$.

### 3. Robust Whitening

The conventional whitening process utilizes the zero time delay correlation matrix of the observed signal, which cannot eliminate the effect of additive noise. In [31,32], a robust whitening algorithm is proposed using the property that the time delay correlation matrix is insensitive to white noise. The time delay correlation matrix of the observed signal is

$$\boldsymbol{R}_x(\tau) = E\left[\boldsymbol{x}(t)\boldsymbol{x}^H(t-\tau)\right] = \boldsymbol{A}\boldsymbol{R}_s(\tau)\boldsymbol{A}^H, \tau \neq 0 \tag{14}$$

where $\tau$ is the time delay and $\boldsymbol{R}_s(\tau)$ is the time delay correlation matrix of the source signal. Since the signals are all complex, $\tilde{\boldsymbol{R}}_x(\tau)$ should be replaced by the ergodic matrix, i.e.,

$$\tilde{\boldsymbol{R}}_x(\tau) = \frac{1}{2}\left(\boldsymbol{R}_x(\tau) + \boldsymbol{R}_x^H(\tau)\right) \tag{15}$$

However, $\tilde{\boldsymbol{R}}_x(\tau)$ is not a positive definite matrix. The literature [32] demonstrated that when the number of delay-dependent matrices and observed signals is equal, a linear combination of multiple delay-dependent matrices can form a positive definite matrix, i.e.,

$$\boldsymbol{C} = \sum_{m=1}^{M} \beta_m \tilde{\boldsymbol{R}}_x(\tau_m) \tag{16}$$

where $\boldsymbol{\beta} = [\beta_1, \beta_2, \cdots, \beta_M] \in \mathbb{R}^K$ is the real vector, i.e., the coefficients of the time delay correlation matrices. $\tau_m (1 \leq m \leq M)$ is the time delay sequence. Thus, the key to the algorithm lies in how to determine $\boldsymbol{\beta}$. $\tilde{\boldsymbol{R}}_x(\tau_m)$ is an ergodic matrix, so $\boldsymbol{C} = \sum_{m=1}^{M} \beta_m \tilde{\boldsymbol{R}}_x(\tau_m)$ is also an ergodic matrix. The minimum eigenvalue of the positive definite matrix should be greater than 0. A new method of finding $\boldsymbol{\beta}$ is as follows. Let the set $\mathcal{C}$ be all linear combinations of $\tilde{\boldsymbol{R}}_x(\tau_m)$, i.e.,

$$\mathcal{C} = \left\{ \sum_{k=1}^{K} \beta_k \tilde{\boldsymbol{R}}_x(\tau_k) \mid \beta \in \mathbb{R}^K \right\} \tag{17}$$

When $\boldsymbol{C} = \sum_{k=1}^{K} \beta_k \tilde{\boldsymbol{R}}_x(\tau_k)$ is a positive definite matrix, the minimum eigenvalue of $\boldsymbol{C}$ should be no less than a given threshold $\sigma \in \mathbb{R}^+$, i.e., $\mathcal{O}_\sigma^+ \cap \mathcal{C} \neq \varnothing$, where $\mathcal{O}_\sigma^+$ represents a set of minimum eigenvalue matrices. It is equivalent to minimizing the following optimization problem.

$$\min F(\boldsymbol{O}, \boldsymbol{C}) = \| \boldsymbol{O} - \boldsymbol{C} \|, \boldsymbol{O} \in \mathcal{O}_\sigma^+, \boldsymbol{C} \in \mathcal{C} \tag{18}$$

$\mathcal{H} = span\{\tilde{\boldsymbol{R}}_x(\tau_1), \cdots, \tilde{\boldsymbol{R}}_x(\tau_K)\}$ is the subspace of $\tilde{\boldsymbol{R}}_x(\tau_k)$ and $O_\mathcal{H}(\boldsymbol{O})$ is the orthogonal projection operator from $\boldsymbol{O}$ to $\mathcal{H}$. $\mathbb{A}$ is the $M^2 \times K$ tensor, i.e.,

$$\mathbb{A} = \left[vec\left(\tilde{\boldsymbol{R}}_x(\tau_1)\right), \cdots, vec\left(\tilde{\boldsymbol{R}}_x(\tau_K)\right)\right] \tag{19}$$

$O_\mathcal{H}$ is

$$O_\mathcal{H} = \mathbb{A}\left(\mathbb{A}^H \mathbb{A}\right)^{-1} \mathbb{A}^H \tag{20}$$

Initialize $C \in \mathcal{C}$, $\beta = [1, 1, \cdots, 1]$, and calculate the eigenvalue decomposition $[U_C, \Lambda_C]$ of $C$. If the minimum eigenvalue is less than the given threshold $\sigma$, then update $O$ and $C$, respectively, according to the following equation.

$$\Lambda_P = \max(\Lambda_C, \sigma I) \tag{21}$$

$$O = U_C \Lambda_o U_C^H \tag{22}$$

$$C = vec^{-1}(O_{\mathcal{H}}(O)) \tag{23}$$

where $I$ is the unit matrix, until the minimum eigenvalue of $C$ is greater than or equal to the given threshold $\sigma$.

Use $C^*$ to denote the final derived positive definite matrix. The eigenvalue decomposition of $C^*$ is performed to obtain the eigenvalues and eigenvectors, respectively. The $K$ large eigenvalues and the corresponding eigenvectors are selected and denoted as $\Lambda_s = diag(\sigma_1^2, \sigma_2^2, \cdots, \sigma_K^2)$ and $U_s = [u_1, u_2, \cdots, u_K]$, respectively, which leads to the following robust whitening matrix.

$$V = \Lambda_s^{-1/2} U_s^H \tag{24}$$

The whitening signal after robust whitening is expressed as

$$z(t) = V x(t) \tag{25}$$

The above method can not only attenuate the effect of noise, but also achieve dimensionality reduction and reduce the amount of computation.

## 4. Objective Function

At present, the objective function commonly used in CBSS is kurtosis maximization in a complex domain. However, kurtosis needs to be averaged many times, which requires a large amount of computation when there are many snapshots. Therefore, this paper chooses the F-norm in joint diagonalization as the objective function. It only needs to multiply the matrix, which can greatly reduce the running time. For a given set of diagonalizable target matrices $M = \{ M^l, l = 1, \ldots, L \}$, where $M^l$ is a square matrix of $K \times K$, the goal of the F-norm objective function is to solve $W$ by minimizing the following equation

$$J(W) = \sum_{k=1}^{K} \text{off}\left( W M^k W^H \right) \tag{26}$$

where $\text{off}(\cdot)$ is the sum of the F-norms of all non-diagonal elements, expressed as

$$\text{off}(F) \triangleq \parallel F - diag(F) \parallel_F^2 = \sum_{i \neq i} F_{ij}^2 \tag{27}$$

where $F$ is an arbitrary matrix. Obviously, the F-norm cost function is to measure the degree to which the matrix group $W M^k W^H$ deviates from the diagonal matrix to solve the optimal solution. $W = 0$ is a global minimal solution satisfying this cost function if there is no additional constraint. However, $W = 0$ is called a trivial solution. Constrained diagonalization matrix $W$ is a unitary matrix, i.e., $W W^H = I$. Obviously, the diagonalized matrix can avoid a trivial solution. At present, there are four-order cumulants of statistically independent sources and second-order statistics of statistically uncorrelated sources to obtain joint diagonalized matrices. Because the fourth-order cumulant of Gaussian white noise is zero, it has a strong anti-noise ability. Moreover, by whitening and selecting multiple cumulant matrices for joint diagonalization, a very robust solution can be obtained. Therefore, the fourth-order cumulant of the whitening signal is used to obtain the joint

diagonalization matrix group. The (*p*,*q*) element in the fourth-order cumulant matrix $\boldsymbol{Q}_z(\boldsymbol{M})$ of the whitening signal $\boldsymbol{z}(t)$ is

$$[\boldsymbol{Q}_z(\boldsymbol{M})]_{p,q} = \sum_{i=1}^{K} \sum_{o=1}^{K} \boldsymbol{Q}_z(p,q,i,o) m_{i,o} \tag{28}$$

where $\boldsymbol{M}$ is any matrix of $K \times K$ and $m_{i,o}$ is the (*i*,*o*) element of matrix $\boldsymbol{M}$. The fourth-order cumulant of whitening signal $\boldsymbol{z}(t)$ is

$$\boldsymbol{Q}_z(p,q,i,o) = cum\big(z_p(t), z_i^*(t), z_q(t), z_o^*(t)\big), \\ p,q = 1,2,\cdots,K \tag{29}$$

where $z^*$ represents the conjugate of $z$, and $z_q(t)$ is the *i*-th line of whitening signal $\boldsymbol{z}(t)$. $cum(\cdot)$ is the fourth-order cumulant operation

$$\begin{aligned} & cum\big(z_p(t), z_i^*(t), z_q(t), z_o^*(t)\big) \\ = {}& E\big(z_p(t)z_q(t)z_i^*(t)z_o^*(t)\big) - E\big(z_p(t)z_i^*(t)\big)E\big(z_q(t)z_o^*(t)\big) \\ & -E\big(z_p(t)z_o^*(t)\big)E\big(z_q(t)z_i^*(t)\big) - E\big(z_p(t)z_q(t)\big)E\big(z_i^*(t)z_o^*(t)\big) \end{aligned} \tag{30}$$

Obviously, the number of elements of $\boldsymbol{Q}_z(p,q,i,o)$ is $K \times K \times K \times K$. $\boldsymbol{Q}_z$ is constructed as an $K^2 \times K^2$ dimensional matrix $\boldsymbol{H}_z$, where

$$\boldsymbol{H}_z(n_1, n_2) = \boldsymbol{Q}_z(p,q,k,l) \tag{31}$$

where $n_1 = p + (q-1)K$, $n_2 = o + (i-1)K$. The $K^2 \times 1$ eigenvectors corresponding to $K$ maximum eigenvalues $\lambda_k$ of matrix $\boldsymbol{H}_z$ are $\boldsymbol{u}_k (k = 1,2,\cdots K)$, and $\boldsymbol{u}_k$ is the performed matrix operation to obtain $vec^{-1}(\boldsymbol{u}_k)$. By the joint diagonalization of $\boldsymbol{M}^k = \lambda_k vec^{-1}(\boldsymbol{u}_k)$, the unmixing matrix can be obtained.

Unitary matrices can be expressed in the form of the continuous multiplication of Givens matrices and extended to the complex field to express the following complex Givens matrices.

$$\begin{aligned} \boldsymbol{W} &= \boldsymbol{T}_{K-1}\boldsymbol{T}_{K-2}\cdots\boldsymbol{T}_1, \\ \boldsymbol{T}_1 &= \boldsymbol{T}_{1,K}\boldsymbol{T}_{1,K-1}\cdots\boldsymbol{T}_{1,2}, \\ \boldsymbol{T}_2 &= \boldsymbol{T}_{2,K}\boldsymbol{T}_{2,K-1}\cdots\boldsymbol{T}_{2,3}, \cdots \\ \boldsymbol{T}_{K-1} &= \boldsymbol{T}_{K-1,K2} \end{aligned} \tag{32}$$

where

$$\boldsymbol{T}_{\rho,\mu} = \begin{pmatrix} 1 & \cdots & 0 & \cdots & 0 & \cdots & 0 \\ \vdots & \ddots & \vdots & & \vdots & & \vdots \\ 0 & \cdots & ce^{j\varphi_1} & \cdots & -de^{j\varphi_2} & \cdots & 0 \\ \vdots & & \vdots & \ddots & \vdots & & \vdots \\ 0 & \cdots & de^{j\omega_3} & \cdots & ce^{j\varphi_4} & \cdots & 0 \\ \vdots & & \vdots & & \vdots & \ddots & \vdots \\ 0 & \cdots & 0 & \cdots & 0 & \cdots & 1 \end{pmatrix}_{N \times N} \tag{33}$$

where $K$ is the matrix order, i.e., the number of sources. $c = \cos\theta$, $d = \sin\theta$, $\theta \in (0, 2\pi)$ is the rotation angle and satisfies $\theta_1 + \theta_4 = \theta_2 + \theta_3$. Because of the constraint $\theta_1 + \theta_4 = \theta_2 + \theta_3$, one unknown parameter can be reduced. Therefore, the number of unknown parameters of each complex Givens matrix is 4. $\boldsymbol{T}_1$ contains $K-1$ complex Givens matrices, $\boldsymbol{T}_2$ contains $2K$ complex Givens matrices, and so on; $\boldsymbol{T}_{N-1}$ contains 1 complex Givens matrix. Each complex Givens matrix contains 4 unknown parameters, so the number of unknown parameters of $\boldsymbol{W}$ is

$$
\begin{aligned}
(1 + 2 + \cdots + (K-2) + (K-1)) \cdot 4 \\
= \frac{(1+(K-1))(K-1)}{2} \cdot 4 \\
= \frac{K(K-1)}{2} \cdot 4 = 4C_K^2
\end{aligned}
\tag{34}
$$

where C is a combination. If $W$ is not represented by the concatenation of complex Givens matrices, since $W$ is of order $K$ and each element is in complex form with 2 unknown parameters, the number of unknown parameters included is $2K^2$. Therefore, the number of unknown parameters that can be reduced $2K^2 - 4C_K^2 = 2K$. For 4 source signals, $W = T_3 T_2 T_1 = T_{12} T_{13} T_{14} T_{23} T_{24} T_{34}$ can be expressed as follows:

$$
W =
\begin{bmatrix}
\cos\varphi_1 e^{j\varphi_2} & -\sin\varphi_1 e^{j\varphi_3} & 0 & 0 \\
\sin\varphi_1 e^{j\varphi_4} & \cos\varphi_1 e^{j(\varphi_3+\varphi_4-\varphi_2)} & 0 & 0 \\
0 & 0 & 1 & 0 \\
0 & 0 & 0 & 1
\end{bmatrix}
\cdot
\begin{bmatrix}
\cos\varphi_5 e^{j\varphi_6} & 0 & -\sin\varphi_5 e^{j\varphi_7} & 0 \\
0 & 1 & 0 & 0 \\
\sin\varphi_5 e^{j\varphi_8} & 0 & \cos\varphi_5 e^{j(\varphi_7+\varphi_8-\varphi_6)} & 0 \\
0 & 0 & 0 & 1
\end{bmatrix}
\cdot
$$
$$
\begin{bmatrix}
\cos\varphi_9 e^{j\varphi_{10}} & 0 & 0 & -\sin\varphi_9 e^{j\varphi_{11}} \\
0 & 1 & 0 & 0 \\
0 & 0 & 1 & 0 \\
\sin\varphi_9 e^{j\varphi_{12}} & 0 & 0 & \cos\varphi_9 e^{j(\varphi_{12}+\varphi_{11}-\varphi_{10})}
\end{bmatrix}
\cdot
\begin{bmatrix}
1 & 0 & 0 & 0 \\
0 & \cos\varphi_{13} e^{j\varphi_{14}} & -\sin\varphi_{13} e^{j\varphi_{15}} & 0 \\
0 & \sin\varphi_{13} e^{j\varphi_{16}} & \cos\varphi_{13} e^{j(\varphi_{16}+\varphi_{15}-\varphi_{14})} & 0 \\
0 & 0 & 0 & 1
\end{bmatrix}
\cdot
$$
$$
\begin{bmatrix}
1 & 0 & 0 & 0 \\
0 & \cos\varphi_{17} e^{j\varphi_{18}} & 0 & -\sin\varphi_{17} e^{j\varphi_{19}} \\
0 & 0 & 1 & 0 \\
0 & \sin\varphi_{17} e^{j\varphi_{20}} & 0 & \cos\varphi_{17} e^{j(\varphi_{20}+\varphi_{19}-\varphi_{18})}
\end{bmatrix}
\cdot
\begin{bmatrix}
1 & 0 & 0 & 0 \\
0 & 1 & 0 & 0 \\
0 & 0 & \cos\varphi_{21} e^{j\varphi_{22}} & -\sin\varphi_{21} e^{j\varphi_{23}} \\
0 & 0 & \sin\varphi_{21} e^{j\varphi_{24}} & \cos\varphi_{21} e^{j(\varphi_{24}+\varphi_{23}-\varphi_{22})}
\end{bmatrix}
\tag{35}
$$

Based on the analysis above, the proposed algorithm converts the solution into the rotation angle in the complex Givens matrix. This algorithm not only guarantees the orthogonality of the separation matrix in the complex domain and enhances the accuracy of the solution, but also reduces the number of unknown parameters. After obtaining the unmixing matrix $W$, the separated signal is

$$
y(t) = Wz(t)
\tag{36}
$$

The separated matrix is

$$
B = WV
\tag{37}
$$

The estimated mixing matrix is

$$
\hat{A} = \left(B^H B\right)^{-1} B^H
\tag{38}
$$

## 5. Optimization Algorithm

Traditional CVBSS commonly utilizes Newton's method or the gradient method as the optimization algorithm. However, these methods are sensitive to the initial value and can be greatly affected by the step size, often resulting in solutions becoming trapped in saddle points. On the other hand, swarm intelligence algorithms offer several advantages, such as high accuracy, fast convergence, and no need for parameter adjustment. In this study, we employ the grey wolf optimizer (GWO) [33,34] as the optimization algorithm. Several GWO algorithms have been developed to improve its performance. Ref. [35] introduces chaos theory into the GWO algorithm to enhance its global convergence. Ref. [36] enhances the global search ability by improving the chaotic tent map used to initialize the wolf, and also uses a nonlinear convergence factor based on the Gaussian distribution curve to balance global and local searchability. The detailed steps of GWO are not extensively discussed. We introduce a new nonlinear convergence factor and chaotic perturbation to enhance the accuracy and convergence speed of the optimization process. This modified algorithm is referred to as the nonlinear chaotic grey wolf optimizer (NCGWO). By incorporating chaotic motion, which possesses spatial ergodicity and extrinsic randomness, the NCGWO

is capable of exploring the entire solution space without repetition. Consequently, the overall quality-seeking capacity and the ability to find local superiority are significantly improved [37].

In GWO, the convergence factor $\rho$ decreases linearly with the increase in iteration times. It is impossible to balance the global and local search capabilities, which is not conducive to the solution of the algorithm. In this paper, a new nonlinear convergence factor is used to improve the GWO. The expression of the convergence factor equation is

$$\rho = \exp(iter - ((Max\_iter + gen)/(Max\_iter - iter))) \tag{39}$$

where *Max_iter* is the maximum number of iterations, and *iter* is the current iteration.

The equation of the Statistical Parametric Map (SPM) is as follows:

$$\gamma(\vartheta+1) = \begin{cases} \mod\left(\frac{\gamma(\vartheta)}{\eta} + \mu\sin(\pi\gamma(\vartheta)) + rand, 1\right), 0 \leq \gamma(\vartheta) < \eta \\ \mod\left(\frac{\gamma(\vartheta)/\eta}{0.5-\eta} + \mu\sin(\pi\gamma(\vartheta)) + rand, 1\right), \eta \leq \gamma(\vartheta) < 0.5 \\ \mod\left(\frac{(1-\gamma(\vartheta))/\eta}{0.5-\eta} + \mu\sin(\pi(1 - \gamma(\vartheta))) + rand, 1\right), 0.5 \leq \gamma(\vartheta) < 1 - \eta \\ \mod\left(\frac{(1-\gamma(\vartheta))}{\eta} + \mu\sin(\pi(1 - \gamma(\vartheta))) + rand, 1\right), 1 - \eta \leq \gamma(\vartheta) < 1 \end{cases} \tag{40}$$

where $\eta$ and $\mu$ are the control parameters, respectively. $\gamma$ is the sequence value of the SPM map. $\vartheta$ is the sequence number. *rand* is a random number. When $\eta \in (0,1)$, $\mu \in (0,1)$, the system is in a chaotic state. As shown in Figure 3, the SPM map is uniformly distributed and has a uniform probability density distribution. It indicates that the SPM map has good ergodicity. The following are the steps of the chaotic perturbation operator.

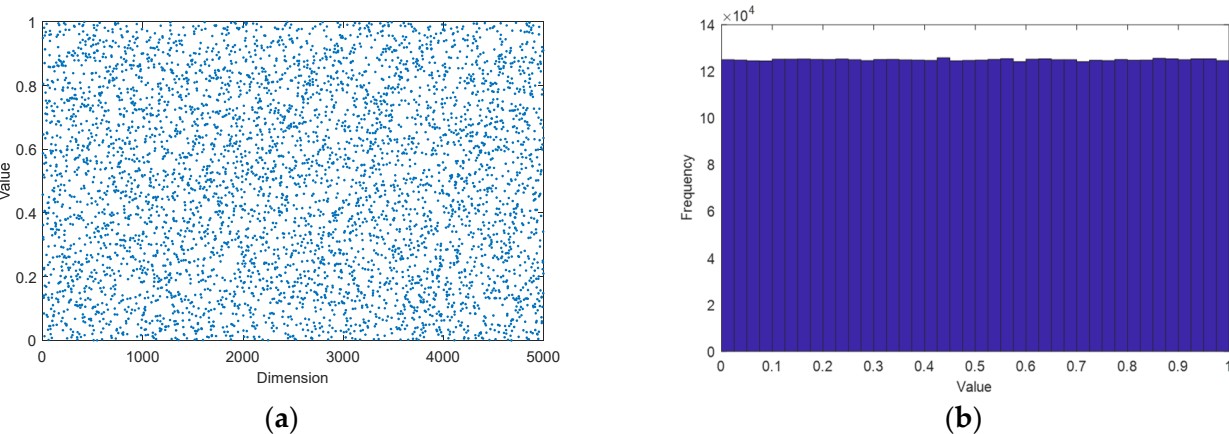

**(a)** **(b)**

**Figure 3.** SPM map, (**a**) Scatter plot, (**b**) frequency statistics.

Step 1: Set the number of chaotic perturbations $R$ and iterate to generate a sufficient number of chaotic sequences. Randomly select $R$ values in the above chaotic sequence to obtain the chaotic sequence $\hat{C} = \{\hat{c}_1, \hat{c}_2, \cdots, \hat{c}_R\}$.

Step 2: Let $\{X_h, h = 1, 2, \cdots H\}$ represent the current wolves, $X_h = \{X_{h1}, X_{h2}, \cdots, X_{hu}, \cdots, X_{hU}\}$ represent the *i*-th wolf, $U = 4C_K^2$. $gX = \{gX_1^*, gX_2^*, \cdots, gX_u^*, \cdots, gX_U^*\}$ represents the position where the optimal wolf is located. After chaotic perturbation $R$ times, $R$ chaotic wolves are obtained within the neighborhood of the optimal wolf, denoted as $\{gX_r, r = 1, 2, \cdots R\}$. Equation (41) represents the position of each wolf.

$$gX_r = \{gX_1^r, gX_2^r, \cdots, gX_u^r, \cdots, gX_U^r\} \tag{41}$$

$$gX_u^r = gX_u^* + \omega \cdot \hat{c}_r \cdot \gamma_u \tag{42}$$

In the above equation, the value in the *j*-th dimensional variable of the *k*-th wolf obtained after chaotic perturbation is denoted by $gX_u^r$, and the perturbation radius in the *j*-th dimensional variable space is denoted by $\gamma_u$. The sign function $\omega$ is obtained according to Equation (43).

$$\omega = \left\{ \begin{array}{ll} 1, & \text{if } rand \geq 1/2 \\ -1, & \text{if } rand < 1/2 \end{array} \right. \tag{43}$$

where $rand \in (0, 1)$.

Each variable space has different ranges, i.e., each variable takes values in different ranges, and the perturbation radius needs to be discussed in dimensions. The perturbation radius of the *j*-th dimensional variable space can be calculated according to the following equation.

$$\gamma_u = \left| \frac{1}{H} \sum_{i=1}^{H} x_{hu} - gx_u^* \right| \tag{44}$$

Step 3: The newly generated *R* chaotic wolves are ranked in fitness with the current *H* wolves, and greedy decisions are made to select the optimal *H* wolves among $R + H$ wolves and proceed to the next iteration.

Algorithm 1 is major steps of the proposed method.

---

**Algorithm 1.** Flow of the proposed algorithm.

---

Input: Observed signal $x(t)$, threshold $\sigma$, chaotic perturbations *R*.

Output: Mixing matrix of estimation $\hat{A}$, separated signal $y(t)$, estimated DOA, BER, estimated target distance and velocity.

Step 1: Wavelet denoising of the observed signal;

Step 2: Calculate the time delay correlation matrix group $\tilde{R}_x(\tau)$

Step 3: Calculate optimal coefficient $\beta^*$ according to Formulas (26)–(28), and get unitary matrix $C^*$;

Step 4: Robust whitening according to Equations (14)–(17); robust whitening is performed according to Equations (29) and (30) to obtain whitening signal $z(t)$;

Step 5: Calculate diagonalizable target matrix group M = $\{M^l, l = 1, \ldots, L\}$ of whitening signals;

Step 6: According to the number of sources, determine the number and dimension of wolves, and initialize the positions of wolves;

Step 7: Calculate the nonlinear convergence factor $\rho$ according to Equation (39) and update the wolves' positions according to (40);

Step 8: Chaos disturbance and greedy decision according to Equations (40)–(44);

Step 9: Judge whether the maximum iteration is reached. If so, output the unmixing matrix and the estimated mixing matrix according to Equations (36)–(38), calculate the separated signal. If not, return to Step 7 for the next iteration;

Step 10: Perform DOA estimation, BER detection, target distance and velocity estimation.

---

## 6. Simulation Analysis

In this paper, the similarity coefficient and performance index (PI) are utilized to quantify the separation performance, while the mean square error (RMSE) is employed to measure the accuracy of direction of arrival (DOA) estimation. The equation is presented as follows.

$$\zeta_{ij} = \zeta(s_i, y_j) = \left| \frac{\sum\limits_{t=1}^{N} y_j(t)s_i(t)}{\sqrt{\sum\limits_{t=1}^{N} y_j^2(t) \sum\limits_{t=1}^{N} s_i^2(t)}} \right| \tag{45}$$

$$PI = \frac{1}{m(m-1)} \left[ \sum_{i=1}^{n} \left( \sum_{j=1}^{n} \frac{|g_{ij}|}{\max\limits_{k}|g_{ik}|} - 1 \right) + \left( \sum_{j=1}^{n} \frac{|g_{ij}|}{\max\limits_{k}|g_{kj}|} - 1 \right) \right] \tag{46}$$

$$\text{RMSE} = \sqrt{\frac{1}{K}\sum_{j=1}^{K}\left(\hat{\theta}_j - \theta_j\right)^2} \tag{47}$$

The true and estimated values of the *i*-th source signal are $s_i$ and $y_j$, respectively. The closer the similarity coefficient is to 1, the more similar the source signal is to the separated signal, and the higher the separation accuracy. ***G*** is the system matrix, ***G* = *BA***, where ***B*** is the separated matrix and ***A*** is the mixing matrix. The *ij*-th element of ***G*** is denoted as $g_{ij}$; the maximum element value of the *i*-th row vector in this matrix is denoted as $\max_k|g_{ik}|$. Similarly, $\left|\max_k|g_{kj}|\right|$ represents the maximum element value of the *j*-th column vector. $\theta_j$ represents the true DOA value, $\hat{\theta}_j$ represents the estimated value, and *K* is the number of DOAs.

### 6.1. Parameter Setting

It is assumed that the enemy radar signal, our detection signal, the jamming signal, and the communication signal are time–frequency mixing. After the mixing signals are separated, DOA estimation, anti-main lobe jamming, target range and velocity estimation, and BER detection are carried out. The target is located at the 2000th range bin, the radial velocity is 60 m/s, and the angle is 60°. The radar transmits a linear frequency modulation (LFM) pulse signal. The bandwidth is 10MHz, the time width is 10 us, and the pulse repetition period is 100us. The jamming signal is sliced jamming of smart jamming, located within the main lobe, and the angle is 58°. The communication signal is quadrature phase shift keying (QPSK) with angle 80°, the carrier frequency is 5 MHz, and the code rate is 2 M/s. The enemy radar transmits a linear FM continuous wave signal. The starting frequency is 0 MHz, the cutoff frequency is 15 MHz, and the angle is 20°. The sampling rate of the above signal is 100MHz, and the observed time is 500 us. The SNR varies from 0 to 20 dB, and the JSR varies from 10 to 30 dB. The radio frequency has been changed to intermediate frequency. Array element spacing is half of the minimum wavelength of the signal and the number of array elements is 20.

### 6.2. Separation Performance Analysis

Figure 4 shows the time and time–frequency plots of the source signal, observed signal (first four channels), and separated signal at SNR = 10 dB and JSR = 20 dB. The waveform plots of 40–60 us are taken for a clear display. From the source signal time–frequency diagram, it can be seen that the source signals overlap each other in the time–frequency domain, which is difficult to be separated by the traditional filtering method. Since the jamming is relatively large, the observed signal has been basically covered by the jamming signal, and only the time–frequency points of the jamming model can be seen.

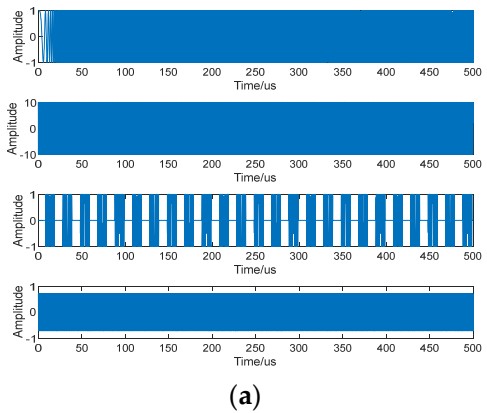

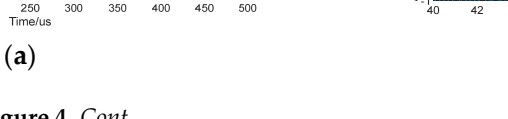

**Figure 4.** *Cont.*

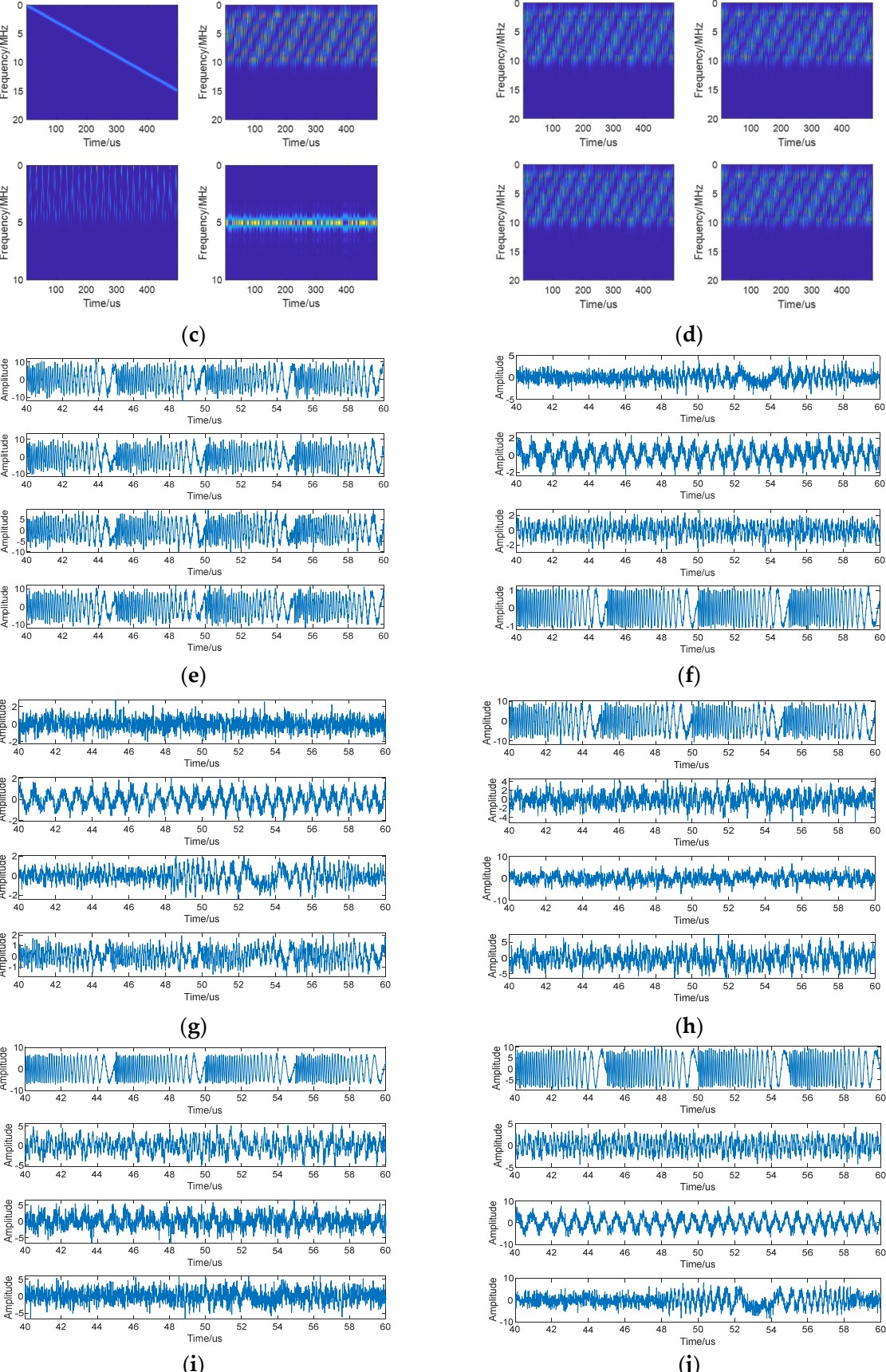

**Figure 4.** *Cont.*

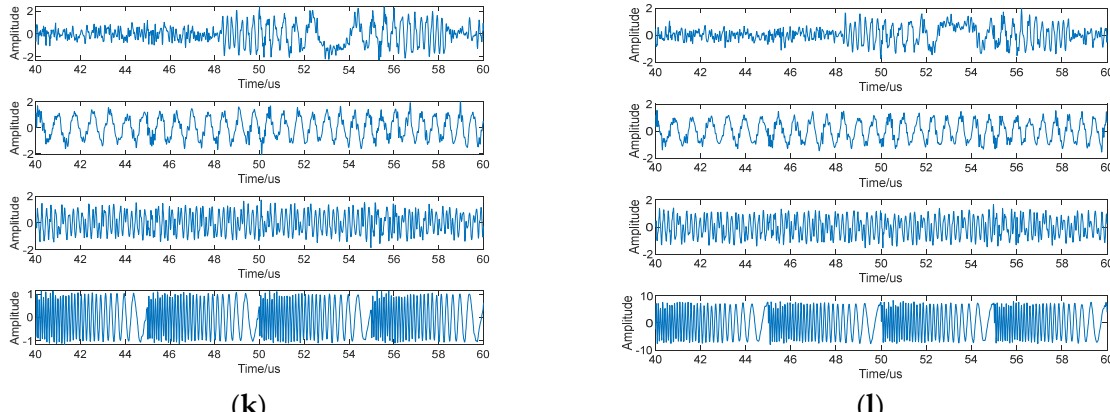

**Figure 4.** Time domain and time–frequency domain plots of the source signal, observed signal (first four channels), and separated signal at SNR = 10 dB and JSR = 20 dB. (**a**) Source signal waveform (0–500 us), (**b**) source signal waveform (40–60 us), (**c**) source signal time–frequency diagram, (**d**) observed signal time–frequency diagram, (**e**) observed signal waveform, (**f**) separated signal waveform (JADE), (**g**) separated signal waveform (cFastICA), (**h**) separated signal waveform (SOBI), (**i**) separated signal waveform (RobustICA), (**j**) separated signal waveform (FAGD), (**k**) separated signal waveform (proposed), (**l**) separated signal after amplitude phase correction (proposed).

The separated signal waveforms clearly show that each algorithm produces different distortions. Due to its high jamming power, the jamming signal separated by each algorithm is the most distinct. Among the six algorithms, the proposed algorithm achieves the highest separation accuracy. It closely resembles the source signal with minimal distortion. After applying amplitude and phase correction, the resulting signal matches the source signal in terms of both amplitude and phase. Table 1 presents the similarity coefficient, performance index, and iteration times for the different algorithms. This algorithm yields the highest similarity coefficients for all signals except the jamming signal. Moreover, the average similarity coefficient exceeds 0.9, indicating a high level of separation accuracy. Additionally, this algorithm exhibits the smallest performance index, suggesting the highest estimation accuracy for the mixing matrix. Compared to cFastICA, RobustICA, and FAGD, it requires the fewest iterations and achieves a faster convergence speed.

**Table 1.** Similarity coefficient, performance index, and iterations.

| Algorithm | JADE | cFastICA | Sobi | RobustICA | FAGD | Proposed |
|---|---|---|---|---|---|---|
| Similarity coefficient | 0.7694 | 0.4318 | 0.5435 | 0.5173 | 0.8384 | 0.8680 |
| | 0.9908 | 0.9701 | 0.9679 | 0.9907 | 0.9796 | 0.9888 |
| | 0.5669 | 0.2867 | 0.2965 | 0.3210 | 0.7079 | 0.8860 |
| | 0.7784 | 0.5428 | 0.4567 | 0.7001 | 0.8311 | 0.9260 |
| Average | 0.7768 | 0.5630 | 0.5662 | 0.6323 | 0.8392 | 0.9160 |
| Performance index | 0.0175 | 0.1281 | 0.1301 | 0.0886 | 0.0086 | 0.0054 |
| Iteration | \ | 55 | \ | 114 | 94 | 38 |

Figure 5 is the objective function curve of Equation (26) obtained by 30 times Monte Carlo with different algorithms. The CGWO [35], IGWO [36], WOA [38], SOA [39], NGO [40], PSO [41,42], SSA [43], and DBO [44] are all excellent swarm intelligence optimizers proposed in recent years. The above optimizers all have the problems of prematurity and low accuracy in different degrees. Compared with GWO, IGWO, and CGWO, NCGWO demonstrates higher convergence speeds and solution accuracy by incorporating improved strategies such as a nonlinear convergence factor and chaotic disturbance strategy. The integration of these strategies enables the NCGWO to generate more dominant solutions,

surpassing the performance of the aforementioned optimizers. Compared with all the above algorithms, the NCGWO proposed in this paper has high accuracy and a fast convergence speed, making it very suitable to solve CVBSS problems.

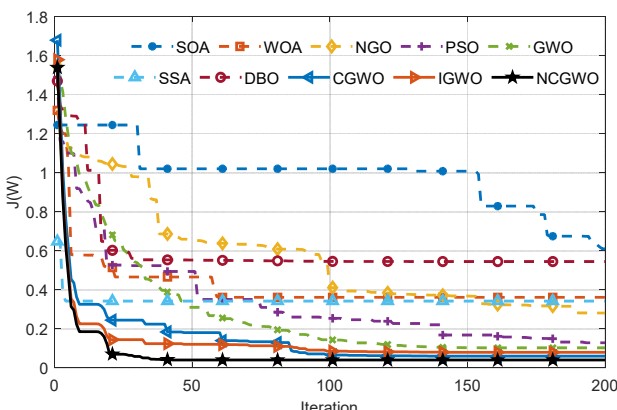

**Figure 5.** Objective function curves of different optimizers.

Figure 6 presents the average similarity coefficients under different SNRs and JSRs. The comparison diagram of similarity coefficients under different SNRs reveals the following observations. (1) As the SNR increases, the separation performance of each algorithm improves. The proposed algorithm exhibits a relatively slow deterioration in performance as the SNR decreases. Notably, when the SNR is between 0 and 10 dB, the proposed algorithm outperforms other algorithms, indicating better separation performance at a low SNR. (2) The performance order of each algorithm is as follows: Sobi < cFastICA < RobustICA < JADE < FAGD < Proposed. The similarity coefficient diagram of different JSRs shows the following trends. (1) As the JSR increases, the separation performance of each algorithm decreases. However, the proposed algorithm exhibits a relatively slow deterioration in performance as the JSR increases. (2) When the JSR is between 20 and 30 dB, the proposed algorithm again outperforms other algorithms, indicating better separation performance under a high JSR. (3) The performance order of each algorithm remains the same: Sobi < cFastICA < RobustICA < JADE < FAGD < Proposed.

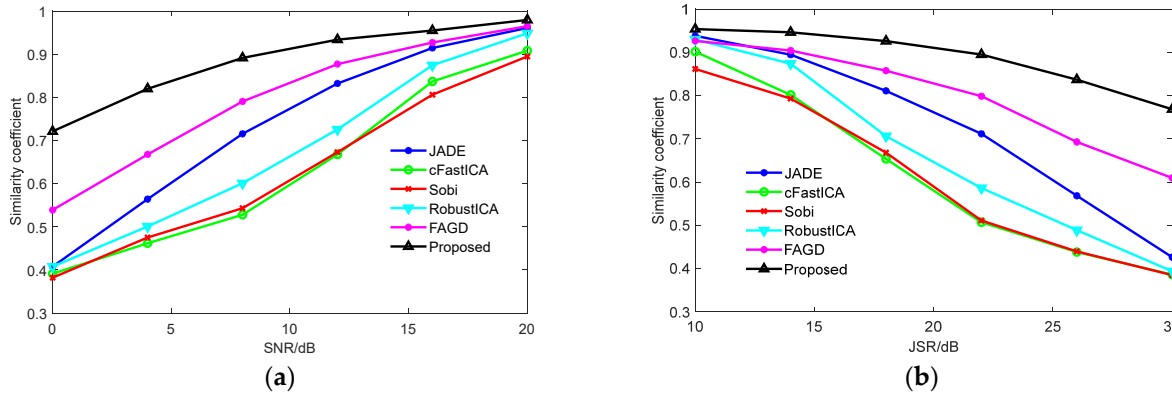

**Figure 6.** Average similarity coefficients under different SNR and JSR. (**a**) Average similarity coefficient under different SNR, (**b**) average similarity coefficient under different JSR.

Figure 7a shows the mean square error of DOA estimation under different SNRs. With the increase in SNR, the DOA estimation RMSE decreases. Obviously, compared with traditional methods, the proposed algorithm is closer to the real value and has higher estimation accuracy. Figure 7b is the BER under different SNRs after coherently demodulating the separated QPSK signal. When the SNR is greater than 10 dB, the BER is 0, so the BER of 0–10 dB is given in the figure. With the increase in SNR, the BER decreases.

Because of the high separation accuracy of the proposed algorithm, the BER under the same SNR is lower than that of other algorithms.

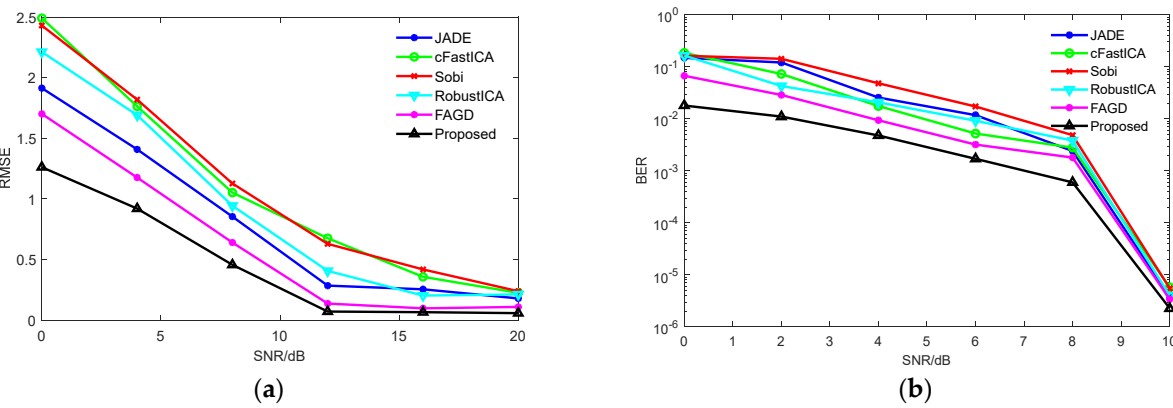

(**a**)

(**b**)

**Figure 7.** DOA estimation RMSE and BER under different SNR. (**a**) DOA estimation RMSE under different SNR, (**b**) BER under different SNR.

### 6.3. Target Range and Velocity Estimation

The integrated system needs to accomplish the detection task, which requires the estimation of the target distance and velocity. The JADE algorithm in [17,18,20] is selected for comparison. Let SNR = 0 dB (array level) and JSR = 30 dB. If pulse compression is performed directly on the observed signal, as shown in Figure 8, multiple blurred false targets will appear and the distance of the target cannot be obtained.

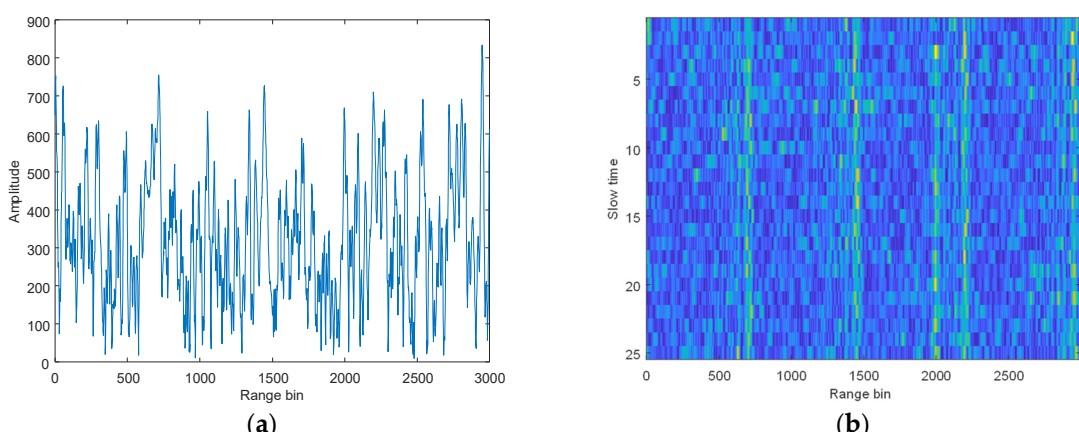

(**a**)

(**b**)

**Figure 8.** Pulse compression of the observed signal. (**a**) Single pulse, (**b**) multiple pulses.

The observed signal is separated using the JADE and pulse compression is performed on the radar echo signal. As shown in Figure 9, multiple spikes appear and the distance at the first spike is not the true distance of the target, resulting in incorrect target detection. After performing pulse compression on multiple pulses, multiple blurred points appear and the distance of the target cannot be estimated. This is caused by the low separation accuracy of the JADE under a low SNR and high JSR.

Figure 10 shows the pulse compression for single and multiple pulses after the separation of the observed signals by the proposed algorithm. Pulse compression of a single pulse shows that the first spike is approximately 700 higher in amplitude than the second spike, and the target distance can be successfully detected. After compressing multiple pulses, a clearer line can be seen, and thus the target distance can be estimated.

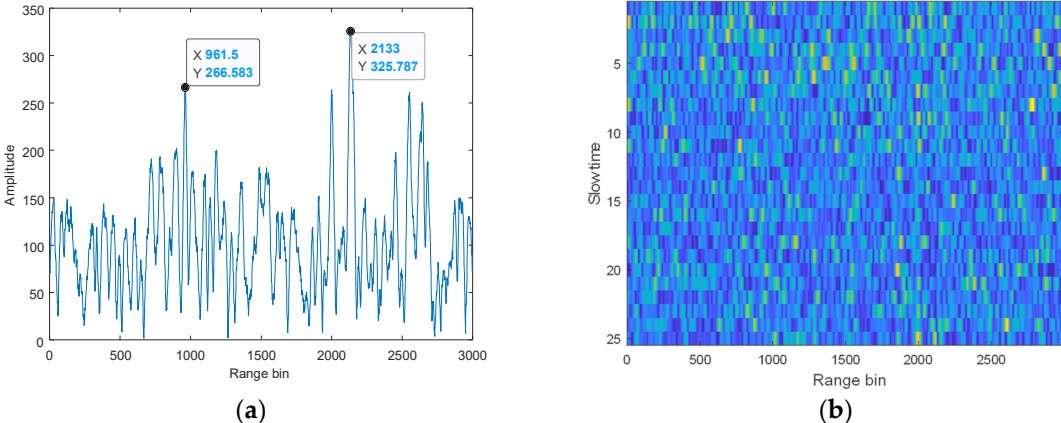

**Figure 9.** Pulse compression of the observed signal (JADE). (**a**) Single pulse, (**b**) multiple pulses.

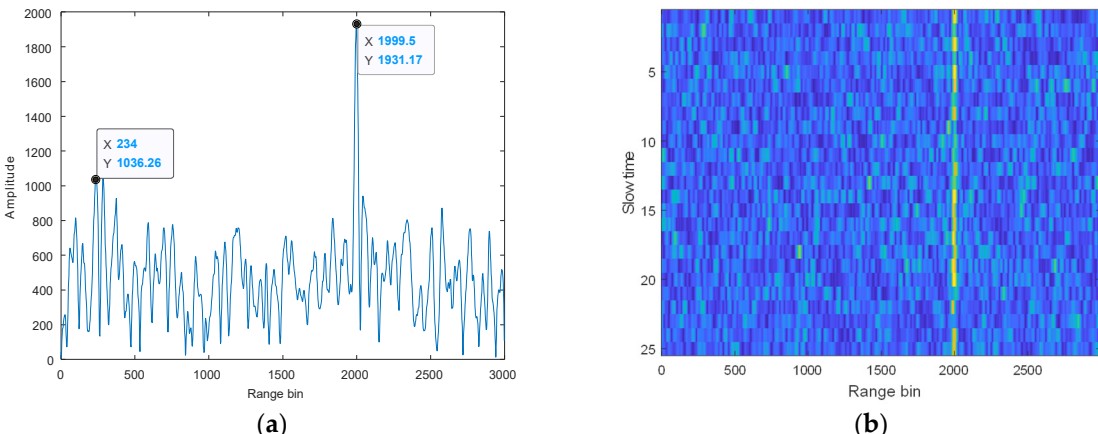

**Figure 10.** Pulse compression of the observed signal (proposed). (**a**) Single pulse, (**b**) multiple pulses.

Figure 11 shows the three-dimensional and two-dimensional plots of the pulse Doppler processing of the radar echo signal, using the proposed algorithm. A peak and a bright spot can be seen in the three-dimensional and the two-dimensional plot, respectively. The proposed algorithm can successfully estimate the target distance and velocity.

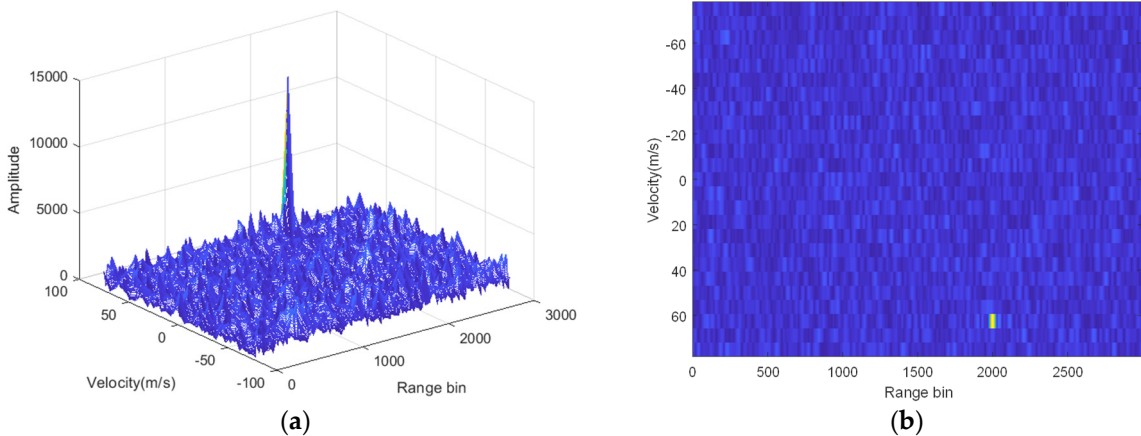

**Figure 11.** Pulsed Doppler processing of the echo signal. (**a**) Three dimensions, (**b**) two dimensions.

## 7. Conclusions

In this paper, a new method for CVBSS is proposed to address the problems of low separation accuracy, slow convergence, and poor robustness in the case of a low SNR and

high JSR. Firstly, a time delay mixing model based on a uniform line array is established, and an amplitude phase correction and DOA estimation method is given on the basis of this model. Then, the whitening matrix is calculated by the linear combination of multiple time delay correlation matrices based on the feature that the time delay correlation matrix is insensitive to noise. Secondly, the whitening matrix is calculated by the linear combination of multiple time delay correlation matrices based on the feature that the time delay correlation matrix is insensitive to noise. Thirdly, to further improve the optimization accuracy and convergence speed, a nonlinear convergence factor and chaotic perturbation operator are added to the GWO. Finally, NCGWO is used to optimize the joint diagonalization objective function. The solution of the separation matrix is transformed into the solution of the rotation angle in a complex-valued Givens matrix. Simulation results show that the proposed algorithm has higher separation accuracy and fewer iteration times compared with the traditional algorithm. At the same time, it can improve the DOA estimation accuracy, reduce the communication bit error rate, and achieve the joint estimation of the target distance and velocity.

Due to equipment limitations and objective conditions, it is sometimes impossible to deploy an extensive number of sensors, resulting in a situation where the number of sensors is smaller than the source signal. BSS with fewer sensors than the source signal is called underdetermined blind source separation. Consequently, the problem of BSS with integrated reception under an underdetermined case emerges as a significant research area for the future.

**Author Contributions:** Conceptualization, W.L. and R.Y.; methodology, X.L.; software, W.L. and K.L.; validation, W.L., H.J. and X.L.; formal analysis, H.L.; investigation, W.L.; resources, H.J. and X.L.; data curation, K.L.; writing—original draft preparation, W.L.; writing—review and editing, W.L. and R.Y.; visualization, K.L.; supervision, H.J.; project administration, R.Y.; funding acquisition, X.L. and H.L. All authors have read and agreed to the published version of the manuscript.

**Funding:** This paper is supported under the National Defense Science and Technology Innovation Zone Fund of China (Grant No. 17H86304ZT00302) and the National Natural Science Foundation of China (Grant No. 61502522).

**Data Availability Statement:** The data that support the findings of this study are available from the corresponding author upon reasonable request.

**Acknowledgments:** The authors would like to acknowledge the National Defense Science and Technology Innovation Zone Fund of China (Grant No. 17H86304ZT00302) and National Natural Science Foundation of China (Grant No. 61502522).

**Conflicts of Interest:** The authors declare no conflict of interest.

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
