# Peer review of "A Novel Complex-Valued Blind Source Separation and Its Applications in Integrated Reception"

_electronics, doi:10.3390/electronics12183954_

Round 1
Reviewer 1 Report
This paper proposes a modified complex valued blind source separation (CVBSS) algorithm for integrated reception cases in difficult conditions with high jamming and low signal-to-noise ratios. The new algorithm improves the estimation accuracy, and reduces the communication errors. This is a good paper with versatile results. Comments:
References: (1) Avoid using lumped references, e.g. line 34 “[1-9]”. Are they all necessary to show that there is a method called Blind Source Separation? It would be better to cite one of the original early articles, and if these papers have some special contribution for this paper, refer to them in more details.
Introduction: (2) The second contribution (lines 81-4) is a little unclear. Do the authors mean “Secondly, because the time delay correlation matrix of observed signals is insensitive to noise, the whitening matrix can be constructed as the linear combination of multiple delay correlation matrices”? (3) The sentence “nonlinear convergence factor and chaotic perturbation is added to the standard Grey Wolf Optimizer (GWO)” is not necessary. See my discussion on this topic later on the next paragraph.
(4) What is the novelty in the “Nonlinear Chaotic Grey Wolf Optimizer” (line 298)? In Section 5, the authors refer to several Swarm Optimizers. but seem to forget that there are several versions of Grey Wolf Optimizers that include chaotic factors and even nonlinear adjustment. How their new algorithm compares to these? Add this discussion in Section 5.
(5) The authors should consider the caption of Fig. 2 once again. Now, there are eight words in some order, which says nothing. Some panels in Fig. 4 are impossible to see (or may have nothing to show).
(6) Conclusion needs a couple of sentences on further research directions on this field.
(7) References. In most cases, the references are written as “Surname, the first letter of the given name” (e.g. #28 Novey M and Adali T.). There are some exceptions that should be corrected: numbers 26, 46 and 47.
Some minor corrections are needed.
Reviewer 2 Report
The simulation results show that proposed algorithm has higher separation accuracy and fewer iterations than the traditional algorithm. Further, it can improve the direction of arrival (DOA) estimation accuracy, reduce the communication bit error rate, and achieve the joint estimation of target distance and velocity under powerful jamming and low SNR. Some comments given to authors as follows:
1. Line 26, The novelty in the current article by the authors is too weak. The past has seen extensive published work of written material. It is required to provide more details for more explanation about the present novel in the introductory section.
2. Line 32-33, some detail justification of low signal-to-noise ratio (SNR) and high jamming-to-signal ratio (JSR) would be discussed.
3. Line 36, not any other option than BSS?
4. Line 90-91, the SWOT analysis for Nonlinear Chaotic Grey Wolf Optimizer (NCGWO) encouraged to provided.
5. Line 95-102, this section feels unnecessary and recommended to deleted.
6. Line 109, please explain the basis of this equation first.
7. Since the present study incorporating numerical simulation of computer program. Please explain the urgency of computational simulation. It brings several advantages compared to experimental and analytical study, such as lower cost and faster results. For this purpose, please provide the information along with relevant reference as follows: https://doi.org/10.3390/su142013413, https://doi.org/10.3390/biomedicines11030951, and https://jurnaltribologi.mytribos.org/v33/JT-33-31-38.pdf
-
